# Comment on van den Hurk et al. The Emergence of Japanese Encephalitis Virus in Australia in 2022: Existing Knowledge of Mosquito Vectors. *Viruses* 2022, *14*, 1208

**DOI:** 10.3390/v15020270

**Published:** 2023-01-18

**Authors:** Michelle Nicole Brinkhoff

**Affiliations:** Climate Change and Public Health, Department of Epidemiology and Preventive Medicine, Faculty of Medicine, Nursing and Health Sciences, Monash University, Melbourne, VIC 3800, Australia; mbri0005@student.monash.edu

I read with interest the article “The Emergence of Japanese Encephalitis Virus in Australia in 2022: Existing Knowledge of Mosquito Vectors” [1]. Andrew van den Hurk et al. (June 2022) extensively reviewed existing information on the Japanese Encephalitis virus (JEV) and the recent unprecedented emergence of JEV across areas of Australia to inform surveillance and control programs [1]. Whilst the article commented on La Niña weather systems and temperature variables, interestingly, climate change was not discussed within the sections of the article focused on the survival and dispersal of JEV. I would like to highlight the impact of climate change, particularly the role of global warming, on increased incidences and the wider geographical range of mosquito-borne JEV occurrences.

JEV is a serious mosquito vector-borne viral encephalitis primarily spread in Australia by infected Culex mosquitoes and hypothetically migrating water birds (who are a natural reservoir for JEV) [2,3]. Whilst pigs are a natural reservoir for JEV, pig-to-pig transmission is rare, and it is not spread from pig to human [4]. Outbreaks generally occur during wet conditions, as the Culex mosquito commonly reproduces in temporary water sources such as puddles, open drains, waterways, flood-plain wetlands, and inland lakes [1,2]. Heavy rainfall over the past two years, linked to La Niña, has resulted in extensive flooding and stagnant water across Australia’s eastern coast, providing abundant breeding areas for mosquitoes and new habitats for migrating water birds [1,2,5,6].

Whilst rainfall influences mosquito and water bird population dynamics, they are also dependent on temperature [1,3]. Australia’s long-term warming has offset the usual cooling influence of La Niña [6]. The Australian land surface has increased by approximately 1.4 °C since national records began, with the annual mean temperature rising from 21.4 °C pre-1910 to 22.65 °C in 2020 [7,8]. Mosquitoes require a temperature greater than 17.5 °C to reproduce; therefore, rising temperatures enable the expansion of their geographical range [3,9]. Increased temperatures affect mosquito reproduction, survival, development, behaviour, geographic distribution, habitats, infective period, feeding rate, and overall transmission of JEV [3,9,10,11,12,13]. Changed temperatures also impact the migration of wading birds, a natural reservoir for JEV, presenting further opportunities for JEV distribution into non-endemic areas [2,3].

It is acknowledged that the relationships between JEV-infected Culex mosquitoes and climate change are complex. Temperature changes can restrict the spread of JEV, with prolonged dry periods reducing breeding sites. Where the conditions have become too hot, vector survival and feeding may be inhibited [3,14,15].

Rainfall, elevated temperature, and the resulting humidity are known environmental drivers necessary for mosquito vector survival. Scientific research indicates that these changes over recent decades will continue, with hotter days becoming more frequent and extreme rainfall events more intense, unless a joint international effort is made to reduce climate change drivers [12,16]. Climate change has serious implications for the transmission and geographic spread of JEV. Therefore, in addition to the findings by van den Hurk et al., it is critical to increase research to better understand how climate change affects the incidence and geographical spread of JEV, as well as to promote practical policies to reduce climate-related risks, both locally and globally [9].

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
