# Peer review of "Comment on van den Hurk et al. The Emergence of Japanese Encephalitis Virus in Australia in 2022: Existing Knowledge of Mosquito Vectors. Viruses 2022, 14, 1208"

_viruses, 2023, doi:10.3390/v15020270_

Round 1

Reviewer 1 Report

The comment on the article by van den Hurk et al (Viruses 14,1208) highlights the role climate change may have played in the rapid spread of Japanese encephalitis virus in Eastern Australia during 2022. Overall, the comment is clear and well written. The following comments are minor and for the authors consideration.

1. The mechanism underlying the spread of JEV over such long distances in Australia has still not definitively been proven. Mosquitoes and wild birds may indeed be the means of spread but this remains a hypothesis, as stated in the blog that the author cites. The author should refer to evidence e.g. isolation of virus from key avian species, if available, or the text should reflect the hypothetical status of the claim and consider other possible mechanisms.

2. The reference citation within the text should follow the journals format.

3. The author could also consider the role climate change plays in virus overwintering and emergence in 2023.

Author Response

Good morning,

Thank you for taking the time to review my letter, your feedback was appreciated.

Response One: Advice considered and changes made.

JEV is a serious mosquito vector-borne viral encephalitis primarily spread in Australia by infected Culex mosquitoes and hypothetically migrating water birds (who are a natural reservoir for JEV) [2,3]. Whilst pigs are a natural reservoir for JEV, pig-to-pig transmission is rare, and it is not spread from pig to human [4]. 

References are updated to reflect changes.

Response Two:

The citation and reference format changed to follow the journal format.

Response Three:

The article remains unchanged on this point.

Reviewer 2 Report

Find attached the letter with some minor text editing.

Author Response

Good afternoon,

Thank you for taking the time to read and review my letter.

Recommendations were accepted and updated accordingly.

Kind regards

Michelle
